# Vectors as Sentinels: Rising Temperatures Increase the Risk of *Xylella fastidiosa* Outbreaks

**DOI:** 10.3390/biology11091299

**Published:** 2022-08-31

**Authors:** Pauline Farigoule, Marguerite Chartois, Xavier Mesmin, Maxime Lambert, Jean-Pierre Rossi, Jean-Yves Rasplus, Astrid Cruaud

**Affiliations:** 1CBGP, INRAE, CIRAD, IRD, Institute Agro, University of Montpellier, 34988 Montferrier-sur-Lez, France; 2AgroParisTech, 91120 Palaiseau, France; 3AGAP Institute, INRAE, CIRAD, Institute Agro, University of Montpellier, 20230 San Giuliano, France

**Keywords:** plant health, *Xylella fastidiosa*, *Philaenus spumarius*, climate change, spy insect strategy

## Abstract

**Simple Summary:**

Although global change is expected to modify the threat posed by plant pathogens, not much is known about how a changing climate will affect the epidemiology of generalist vector-borne diseases. In the present study, we developed a high-throughput screening method to test for the presence of a deadly plant pathogen, *Xylella fastidiosa*, in its insect vectors. Based on a four-year survey in climatically distinct areas of the island of Corsica (France), we found a significant positive correlation between the frequency of insect vectors positive for *X. fastidiosa* and temperature. We observed that a higher prevalence in insects corresponded with milder winters. We used future climate projections up to the year 2100, and found that the risk for *X. fastidiosa* outbreak will increase in the future. While the proportion of vectors that carry the pathogen should increase, the climate conditions will remain suitable for the bacterium and its main vector, with possible shifts towards higher elevations. Besides calling for research efforts to limit the incidence of plant diseases in temperate zones, this works reveals that recent molecular technologies could and should be used for massive screening of pathogens in vectors in order to scale-up surveillance and management efforts.

**Abstract:**

Global change is expected to modify the threat posed by pathogens to plants. However, little is known regarding how a changing climate will influence the epidemiology of generalist vector-borne diseases. We developed a high-throughput screening method to test for the presence of a deadly plant pathogen, *Xylella fastidiosa,* in its insect vectors. Then, using data from a four-year survey in climatically distinct areas of Corsica (France), we demonstrated a positive correlation between the proportion of vectors positive to *X. fastidiosa* and temperature. Notably, a higher prevalence corresponded with milder winters. Our projections up to 2100 indicate an increased risk of outbreaks. While the proportion of vectors that carry the pathogen should increase, the climate conditions will remain suitable for the bacterium and its main vector, with possible range shifts towards a higher elevation. Besides calling for research efforts to limit the incidence of plant diseases in the temperate zone, this work reveals that recent molecular technologies could and should be used for massive screening of pathogens in vectors to scale-up surveillance and management efforts.

## 1. Introduction

Increased globalization of food production and climate change are facilitating the movement and local establishment of pests and pathogens [1]. The spread of plant pathogens is jeopardizing food security [2] and biodiversity [3]. Identifying how plants, pathogens, and their vectors will respond to a changing climate is challenging [4], even more so for unmanaged ecosystems [4,5,6]. However, improvements in molecular detection and climate modelling allow us to model realistic scenarios more accurately. Bacterial vector-borne plant diseases represent a major cost to producers worldwide [7], because they decrease the overall yield and require expensive control measures. Bacteria are mainly transmitted to plants by Hemipteran insects (e.g., psyllids, leafhoppers, and spittlebugs) that feed on the mesophyll, phloem, or xylem sap [8]. Climate change is expected to affect the distribution of vectors and, hence, the geographical range over which diseases are transmitted. Ambient temperature is also an important factor for determining the efficiency with which vectors transmit pathogens, how well plants can defend themselves, and possibly the multiplication rate of pathogens within hosts [9]. Despite generalist bacteria posing a severe and costly threat to both managed and unmanaged systems, it is largely unknown how they will respond to climate change.

With about 600 host species in over 85 families of wild and cultivated plants, the vector-borne bacterium *Xylella fastidiosa* Wells et al. (*Xf*) (Xanthomonadaceae) is a worldwide threat to agriculture, horticulture, forestry, and unmanaged habitats [10]. Biofilm-like colonies are formed; pectin gels and tyloses are produced by the plant, which reduce the hydraulic conductivity within the xylem and can lead to plant death [11]. *Xf* is transmitted to plants by xylem-feeding hemipterans (mainly sharpshooters and spittlebugs) [12,13]. There is no vertical transmission to offspring and infectivity is lost during molting, although adults that acquire *Xf* remain infective for life [13]. Sharpshooters can inoculate *Xf* with no latent period [14] and most vectors are polyphagous [12,13], which increases chances of transmission within and between semi-natural and cultivated habitats. At its center of origin in the New World, the bacterium has caused more than 100 million USD worth of losses each year to the Brazilian citrus industry and to the US grape industry [15,16]. Since the dramatic 2013 outbreak in the olive groves of Southern Italy [17], the presence of *Xf* has been confirmed in different Mediterranean regions of Europe (https://gd.eppo.int/taxon/XYLEFA/distribution, accessed on 13 July 2022). The economic impact on the European olive industry over 50 years could reach 5.2 billion EUR (5.6 billions USD) depending on cultivar resistance and effective control of the disease [18]. It is reasonable to expect that climate change will play a role in the worldwide epidemiology of *Xf* [19,20,21,22,23], but too few experimental studies have been conducted so far with which to generate robust predictions [19,20].

We conducted a four-year survey (2016–2019) across a range of climates in Corsica (France; Figure 1 and Figure 2), where *Xf* was detected for the first time in the summer of 2015. We explored relationships between climate variables (temperature and precipitation) and the prevalence of *Xf* in vectors (defined as the proportion of insects that carry *Xf* in sampled populations). We also generated hypotheses under a range of global change scenarios for the future climate suitability of Corsica for *Xf* and its main vector and for the prevalence of *Xf* in vectors. Our results are discussed with consideration regarding what we do, and do not know, about the pathosystem (*Xf*, its vectors, and their host feeding plants).

## 2. Materials and Methods

**Sampling sites and collection of specimens**—The sampling sites (Figure 1, Appendix A) were selected with the primary aim of maximizing the range of climatic conditions (climate data retrieved from SAFRAN models (see next section)). Sampling sites were located in various types of vegetation (Figure 2A; vegetation data retrieved from the OCS GE database (© IGN—2022, https://geoservices.ign.fr/ocsge, accessed on 12 July 2022)) and at varying altitudes (Figure 2B). The sampling sites were not selected based on official detections of *Xf* in plants. Indeed, false positive or negative results may exist and surveillance does not cover all areas because it is almost exclusively directed towards symptomatic plants. We considered repeated measures for some sites to obtain an idea of the steadiness of *Xf* prevalence through time. However, as much as possible and because our aim was not to analyze time series, we selected insect populations originating from different sites to minimize data inter-dependence in the data set. Indeed, as advocated by Colegrave and Ruxton [24], even if data inter-dependence was now correctly accounted for by random effects in the generalized linear mixed model (GLMM) [25], it remains more powerful to collect independent data. A minimum of 30 adults were collected by sweep netting the vegetation at each sampling site (Appendix A). Insects were killed on site with ethyl acetate, and were quickly transferred to 8 mL vials filled with 70% EtOH and stored in 96% EtOH about 10 days later. The vials were stored at 4 °C until DNA extraction. The adults were collected in late October so that they had time to feed on infected plants and *Xf* could multiply in their foregut [26].

**Prevalence data**—Quantitative and conventional PCR are currently established as the gold standard when testing for the presence of *Xf* in plants, but drawbacks regarding sensitivity and versatility have been highlighted for vectors [27]. Therefore, we developed a high-throughput method to amplify and sequence *leuA,* one of the housekeeping genes of *Xf* [28], in insects*. Philaenus spumarius* (L.) (Aphrophoridae), the most significant vector of *Xf* in Europe [12], was used as a sentinel to track *Xf* in the environment. DNA was extracted from single specimens following Cruaud et al. [27]. A two-step PCR approach targeting *leuA* was set up following Cruaud et al. [29]. Details on the protocol are provided in the Appendix A. A unique combination of 9-nt indexes was assigned to each specimen to track index hopping, and four PCR replicates were performed per insect to reduce false negatives (the same combination of indexes for all replicates). Sequencing was performed on a MiSeq system (2 × 250 bp). Analysis of the raw data was adapted from Cruaud et al. [29]. Adapter trimming and the selection of high quality paired reads was performed with Trimmomatic [30]; paired reads were merged with FLASH [31]; clustering of sequences was performed with SWARM [32]. Consensuses were aligned to the set of reference sequences available in pubMLST (http://pubmlst.org/xfastidiosa, accessed on 12 July 2022) and alignment was visually inspected in Geneious R11.1.4 (https://www.geneious.com, accessed on 12 July 2022) to discard non-target amplifications. The complete analytical workflow with examples is available from https://github.com/acruaud/prevalenceXfinsectclimate_2022 (accessed on 12 July 2022) and details are provided in Appendix A. The two step PCR approach was used for specimens collected in 2017–2019. Prevalence data for 2016 were retrieved from a previous study (nested PCR, Sanger sequencing [27]). We note that prevalence data for 2016 were not statistically different from those of the following years (Appendix A).

**Climate variables**—Twenty-five temperature- and five precipitation-related variables were computed to describe the climate profile of the sampling sites. Variables were chosen considering the phenology of *P. spumarius* in Corsica; literature on the multiplication of *Xf* in plants and *P. spumarius* and, in the absence of knowledge on the epidemiological dynamics of *Xf multiplex* in Europe, annual fluctuations of Pierce’s disease incidence in California. Five time slices were defined (as for growing and dormant seasons for plant/insects: March–November, December–February, and regarding *P. spumarius* phenology: March–June, July–August, and September–October). For each time slice, we computed the daily mean temperature. For each time slice but for the dormant season, we computed the maximum temperature of the time slice and the average daily maximum temperature over the time slice. For the dormant season, we computed the minimum temperature of the time slice and the average daily minimum temperature over the time slice. Finally, for the growing season, we computed the number of days with a daily maximal temperature strictly greater than 16 °C, 18 °C, 20 °C, 22 °C, 24 °C, and 30 °C, while the number of days with a daily minimum temperature strictly lower than 0 °C, 2 °C, 4 °C, and 6 °C was computed for the dormant season. For each time slice but for the dormant season, we computed the sum of the daily precipitation. Finally, we computed the sum of the daily precipitation for the growing season of the year Y-1 (see Appendix A for details). Raw data to compute climate variables were retrieved from Météo France (SAFRAN model), which interpolates temperature/precipitation measures made several times a day by a network of over 1000 meteorological stations spread over the French territory. SAFRAN provides the daily temperature (2 m above ground) and precipitation data simulated at a resolution of 8 km on an extended Lambert-II projection that were used to compute the studied climate variables. To estimate future and past climate conditions under different global circulation models (GCMs) and shared socioeconomic pathways (SSPs) (Appendix A), we relied on bioclimatic variables from the CHELSA v2.1 database (https://chelsa-climate.org, accessed on 12 July 2022).

**Statistical analyses**—GLMMs were built to analyze the effect of climate variables on prevalence (binomial distribution). Independent climate variables were built with two methods. First, a principal component analysis (PCA) was performed on the 30 climate variables and the scores of the sampling sites on PC1 and PC2 were used as the input for a first GLMM (GLMM1). Second, a partial least square regression analysis (PLSR) was conducted to rank climate variables in decreasing importance (using the Variable Importance on Projection, VIP [33]) regarding the correlation with *Xf* prevalence. Climate variables were selected step by step, in decreasing VIP order, with two conditions: VIP > 1 and Spearman correlation coefficient with variables selected in the previous steps lower than 0.7 [34]. The climate data table for sampling sites reduced to the selected variables was used as the input for a second GLMM (GLMM2). To account for repeated measures, we added the random effect of the sample site identifier. The year was included as an experimental design fixed effect owing to the number of factor levels being below 5. GLMM validity (correct distribution, dispersion, frequency of outliers, and homoscedasticity) was ascertained and we tested the significance of the fixed variables in models with type II analyses of deviance (two-sided type II Wald chi-square tests) (see Appendix A for all details).

**Species distribution modelling—**Species distribution models (SDMs) were built from worldwide occurrences of *Xf* ssp. *multiplex* and *P. spumarius* using 12 bioclimatic variables (Appendix A; CHELSA database) and the Maxent algorithm [35] with 10,000 background points to define the available environmental conditions. The SDMs were used to predict the species future potential distribution using the same combinations of GCMs and SSPs as those used to model the future climate profiles of the sampling sites (see Appendix A for all details).

## 3. Results

Of the 1200 insects tested for *Xf* (39 populations of *P. spumarius* across Corsica; Appendix A), 8% were recovered as positive for *Xf* ssp. *multiplex*. Prevalence in insect populations ranged from 0 to 40% (Figure 1, Appendix A). The first (PC1) and second (PC2) axes of the PCA performed on the 30 climate variables respectively supported 56.7% and 12.9% of the climate variability. PC1 opposed plots with high temperatures to plots with low temperatures. PC2 opposed plots with high maximal temperatures and high precipitations during the previous year to plots with high precipitations. In both GLMMs, variables that were significantly correlated with *Xf* prevalence were strongly linked to temperature descriptors (Appendix A and Figure 3). GLMM1 (built from scores of the sampling sites on PC1 and PC2 of the PCA; Appendix A) and GLMM2 (built from uncorrelated most explanatory variables according to PLSR) provided similar results, showing that *Xf* prevalence in vectors was positively correlated to temperature. In GLMM1, prevalence decreased with the sampling site score on PC1, meaning that prevalence was higher in sites with higher temperatures (Figure 3A). GLMM2 identified the number of days from December to February, with a minimal temperature <6 °C (d6C_dec_fev) as the most explanatory variable. A higher prevalence was predicted under milder winter (Figure 3B).

To explore the past and future climate profiles of the sampling sites, we computed the mean temperature of the coldest quarter (bio11), which is the bioclimatic variable that is the closest to d6C_dec_fev (Appendix A; Spearman’s rank correlation coefficient between bio11 and d6C_dec_feb for current climate = −0.87)) and for which the past and future values are publicly available. For all combinations of GCMs and SSPs, a general increase in winter temperatures (Figure 4 and Appendix A) was predicted. This, together with GLMM predictions, suggests that milder winters in the future will favour an increase in the prevalence of *Xf* in vectors.

Only uncorrelated bioclimatic variables (variance inflation factor < 10) were retained for SDMs (bio5, bio8, bio9, bio10, bio11, bio13, bio17, and bio19 for *Xf*; bio5, bio6, bio8, bio9, bio13, bio14, and bio18 for *P. spumarius*). The MIAmaxent stepwise model fitting procedure led to a significant model (*p* = 6.83 × 10^−5^) based on four explanatory climate variables for *P. spumarius,* namely: bio5 (variable contribution = 53.7%), bio6 (36.6%), bio8 (4.9%), and bio13 (4.9%). The model was also significant for *Xf* (*p* = 1.11 × 10^−3^) and five variables were retained, namely: bio11 (60.0%), bio5 (24.1%), bio19 (13.0%), bio9 (1.5%), and bio8 (1.5%). Both models exhibited good evaluation metrics. The area under the curve (AUC) values and continuous Boyce indexes (CBI) were 0.822 and 0.992, respectively, for *P. spumarius* and 0.993 and 0.976, respectively, for *Xf*. Figure 5 shows the climate suitability of Corsica in the form of a consensus model based on the median of the predicted climate suitability using the model of each species and the five GCMs. Consensus models were computed for the periods covering 2011–2040 and 2041–2070 for both SSP370 and SSP585 using the medium to high end of plausible future pathways of greenhouse gas emissions (radiative forcing reaching respectively 7.0 W/m^2^ or 8.5 W/m^2^ in 2100; Appendix A). Other results (time periods and SSPs) are available in Appendix A. All predictions suggest that both *Xf* and its main vector will keep encountering suitable climate conditions in Corsica in the future. In the most extreme scenarios, climate change may lead to lower suitability on the coastal areas of Corsica for both partners, but suitable conditions could be found in areas that were previously unsuitable or less suitable, especially in higher elevation areas. The surface corresponding to suitable conditions for both species, i.e., for which there is an overlap between the vector and the pathogen decreases for the most extreme situations (Appendix A, 2071–2100, SSP370 and SSP585), but remains high in all of the scenarios explored.

## 4. Discussion

This study has two major outcomes. First, it highlights the power of insect vectors to track *Xf* in ecosystems. Currently, the surveillance of *Xf* is almost exclusively directed towards symptomatic plants. However, infected vectors have now been recorded twice in areas supposedly free of *Xf* based on plant survey (e.g., northwestern and northeastern Corsica; this study and [27]). These findings clearly indicate the need for alternative early warning and long-term monitoring systems. Indeed, the surveillance of *Xf* in plants is actually more challenging than in insects. First, *Xf* is heterogeneously distributed within the plant tissue [36], which can lead to false negatives. This source of false negatives does not exist with insects for which the whole body is analyzed. In addition, the diversity of PCR inhibitors associated with numerous *Xf* host plant matrices adds technical challenges to the effective detection of the bacterium [37]. Finally, because many species are asymptomatic to low bacterial loads in natural conditions [23], targeting only symptomatic plants is not effective. A nuanced understanding of the factors leading to pathogenicity in this endophytic bacterium requires an explicit inclusion of insect vectors. Insects need to be surveyed at large scale using cutting-edge molecular tools.

The second major outcome is that the prevalence in vectors is highly likely to increase with ambient temperature. It has been established that *Xf* spread is directly linked to the abundance of infectious vectors [22], while plant mortality relates to the number of inoculation events [38]. A corollary of these factors is that areas experiencing milder winters and warmer springs and falls are at greater risk of new outbreaks. A mechanistic understanding requires a summary of what is known concerning the effect of temperature on *Xf* and its main vector.

Regarding the bacterium, the optimal temperature range may vary among strains of *Xf*, but temperatures below 10 °C and above 32 °C affect the survival of the bacterium according to in vitro and potted plant experiments [39]. Within this range, increasing temperatures favor higher multiplication rates [39], with plants achieving an infectious status faster [20]. Overwinter recovery of plants is observed in natural conditions and experiments show that exposure to freezing temperatures can lead to temporary or complete remission of symptoms [40]. Plants (vines) inoculated later in the growing season have better chances of recovery than those inoculated earlier [41]. However, the mechanisms leading to recovery are still not fully understood. Regarding the development of *P. spumarius*, data on the influence of temperature are inconsistent [42]. The minimum temperature for egg hatching and nymphal development ranges from 2.8 °C to 10 °C, and development is still observed at 27 °C, while the nymphal period becomes shorter with increasing temperatures [42]. Summer droughts can negatively impact *P. spumarius* populations and shifts from dry to less water-stressed plants or migration toward cooler climates have been documented [43,44]. In Corsica, field observations suggest that low numbers of adults survive until mid-February [45], while the first nymphs hatch in early February. Adults are virtually impossible to find in the summer, including in the riparian vegetation [45]. As such, the effects of temperature on the probing behavior or feeding rate of *P. spumarius* are unknown. The same applies for *Xf* multiplication within the insect foregut or transmission efficiency. In the US, a positive temperature-dependent transmission efficiency has been highlighted for some vectors [21] and a higher temperature also favors flight activity, feeding, and overwinter survival [22].

Our current understanding of vector and pathogen ecology allows us to propose the following. Milder winters likely increase the overwinter survival of *Xf* in plants, and the bacterial load is consequently higher in spring. The multiplication of *Xf* is favored by warmer weather during the growing season. A high cell density (which promotes biofilm formation [46]) is achieved earlier and, consequently, acquisition by insects happens sooner. A higher temperature in the growing season may also increase vector activity. Vectors may fly more frequently, disperse further, and take longer meals, which could favor the acquisition [47] and transmission of *Xf*. The probability of encounters between vectors and *Xf* is thus steadily increasing. A better understanding of the whole process would require lab and, above all, given the expected discrepancies, field experiments. These field experiments should primarily be designed to follow bacterial load in insects and plants throughout the year, but will also help to document unknown aspects of the ecology of *P. spumarius*. Of particular interest would be studies on overwinter survival of *P. spumarius,* because survival may steadily increase with milder winters. The population size may also increase and lead to a higher transmission efficiency. All of these factors could affect the epidemiology of *Xf* [48]. Obviously, an increased understanding of the environmental factors on other components of this complex pathosystem is also crucial [49,50].

We show that the prevalence might continue to increase, but how do these factors align across the pathosystem? Are vector populations, prevalence in vectors, and plant symptoms all favored by the same climatic conditions? As shown by our SDMs, the effect of climate on vector populations will probably be context-dependent: conditions becoming hotter and drier will favor *P. spumarius* at the cool-moist end of our climatic gradient (i.e., on the most elevated sites), while they will have an adverse effect on *P. spumarius* at the hot-dry end of our gradient (e.g., on sites already experiencing heat waves of up to 37.8 °C in summer, Appendix A). This is in line with the preference of *P. spumarius* for moist environments in Mediterranean climates [42,51] and the preference of vectors for fully irrigated plants [47]. The link between climate and *Xf* prevalence was clear for temperature, but inconclusive for precipitation, as none of the precipitation-related variables were retained in our models. The link between temperature and *Xf* symptoms is understudied, but Pierce’s disease severity is expected to be strongest in the warmest places of the Mediterranean basin, especially in those experiencing mild winters [52]. Finally, a negative feedback loop is likely to operate between vector populations and symptoms, because vectors strongly prefer to feed on healthy or asymptomatic plants in controlled conditions [20]. As a result, it is unlikely that rising temperatures will contribute to an exponential increase in the number of outbreaks. However, we can anticipate (i) increasing *Xf* prevalence and symptoms in the warmest places of Corsica, together with a decrease in *P. spumarius* abundance, and (ii) a progression of the *Xf* pathosystem towards more elevated sites, with the build-up of large vector populations. It is worth noting that adaptation to climate change for both partners (e.g., possibly longer aestivation for the vectors) is unknown and may influence our projections [53]. A final argument against runaway resides in the biology of *Xf* itself, which, for most plants, is a commensal exhibiting self-limiting behavior through quorum sensing [46].

Mitigating the effect of global warming requires knowledge on how the climate may affect different aspects of plant pathosystems [20]. Here, we provide a first assessment of how increasing temperatures may affect the prevalence of *Xf* in vectors. In addition, increased market globalization is also of concern as it may favor the introduction to Europe of other efficient vectors (e.g., *Homalodisca vitripennis* and *Graphocephala atropunctata*) or bacterial strains with which hybridizing is possible with unpredictable outcomes [54].

As illustrated here for *X. fastidiosa*, recent works have suggested that climate change will result in increasing the burden of plant pathogens at a high latitude in the Northern Hemisphere, particularly in Europe, China, and the central to eastern US [2,5]. Impact may vary depending on the ability of natural ecosystems and production systems to adapt [2,5,6]. Preventive and, when possible, curative plant protection have been underlined as key components to maintain and preserve current and future food security [5]. However, managed and unmanaged ecosystem should not be considered as separate compartments [6] with surveillance mainly targeting symptomatic cultivated plants. Indeed, especially for generalist vector-borne plant diseases, genetically diverse wild plants should be seen as potential reservoirs of pathogens for crops [6]. Conversely, infected cultivated plants that are introduced at the vicinity of natural ecosystems could become a source of biodiversity loss. In a rapidly changing world, early warning and long-term monitoring systems are crucial. This study is an example of how and why new sequencing technologies targeting pathogens in vectors are essential for scaling up surveillance efforts and protecting plant health. A blind and massive screening of vectors could also reveal undocumented vector–pathogen associations that may influence disease dynamics.

## 5. Conclusions

This study highlights how the surveillance of *Xylella fastidiosa* could be improved by using insects in addition to symptomatic plants. Indeed, we show that infected vectors are found in localities where symptomatic plants have never been recorded. This survey is an example of how and why new sequencing technologies targeting pathogens in vectors are essential for scaling up surveillance efforts and long-term monitoring.

Our results clearly indicate that *Xf* prevalence in its insect vectors is very likely to increase with ambient temperature. Areas experiencing milder winters and warmer springs and falls are expected to be at greater risk of outbreaks. Projections of the future potential distribution of both *P. spumarius* and *X. fastidiosa* indicate that although the climate may alter their current distribution, both species will find suitable climate conditions in more elevated areas of Corsica.

## Figures and Tables

**Figure 1 biology-11-01299-f001:**
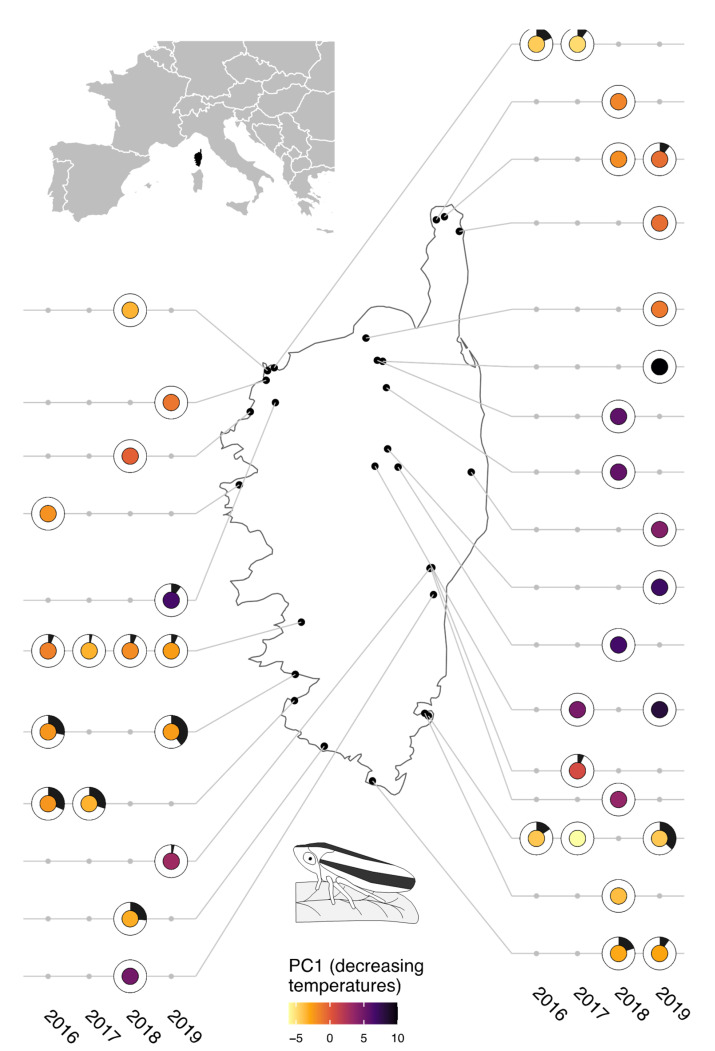
**Prevalence of *X. fastidiosa* in populations of insect vectors.** The black portion of each donut represents the proportion of specimens from which *Xf* was sequenced. The color of the pies shows the score of the sampling sites on PC1 of the PCA performed on climate variables (the warmest colors correspond to the highest temperatures).

**Figure 2 biology-11-01299-f002:**
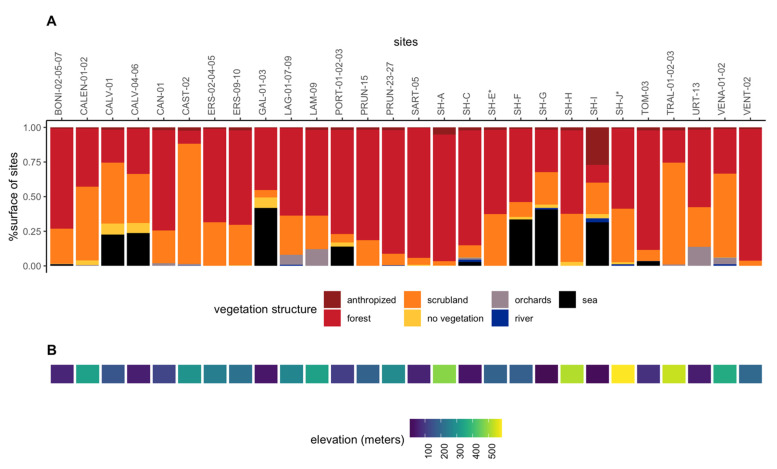
**Vegetation structure and elevation of the sampling sites**. (**A**) Proportion of each vegetation structure in all sites sampled (**B**) Elevation of the sample sites (in the same order as in A). The vegetation structure is given within a radius of 1 km around the sampling sites. See also the additional map.

**Figure 3 biology-11-01299-f003:**
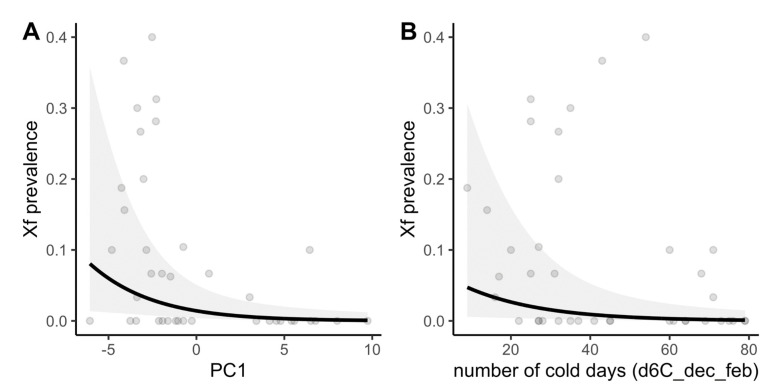
Correlation between *X. fastidiosa* prevalence in vectors and climate. (**A**) GLMM1: Prevalence and sampling site scores on PC1 of the PCA performed on climate variables. (**B**) GLMM2: Prevalence and number of days in December to February (d6C_dec_feb) with minimal daily temperature <6 °C. Points are raw data, lines are regression curves from GLMMs, and grey ribbons are 95% confidence envelopes.

**Figure 4 biology-11-01299-f004:**
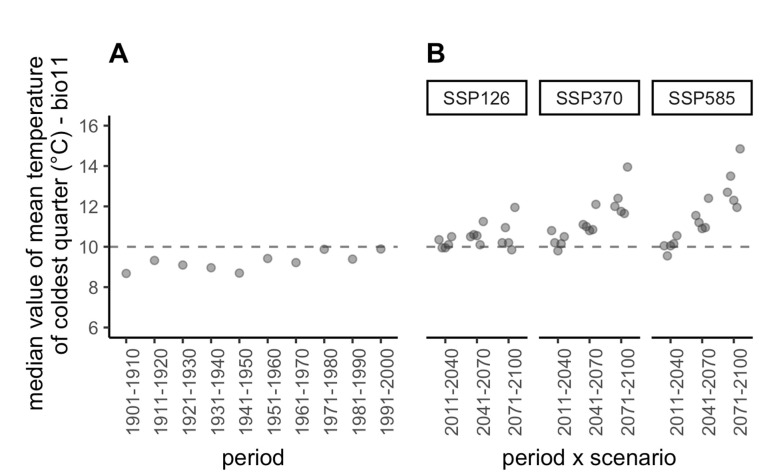
**Past and future projections of temperature for the coldest quarter.** (**A**) Median value of bio11 (mean temperature of coldest quarter) over all sampling sites projected in the past. (**B**) Median value of bio11 over all sampling sites projected in the future under three shared socio-economic pathways (SSPs, three facets) and five global circulation models (GCMs, five points per facet jittered along abscissa) that reflect uncertainties regarding the response of humans to climate change and the evolution of physical processes in the atmosphere, oceans, cryosphere, and land surface, respectively (see Appendix A for details). The dashed line shows the current median value (for the period 2000–2018).

**Figure 5 biology-11-01299-f005:**
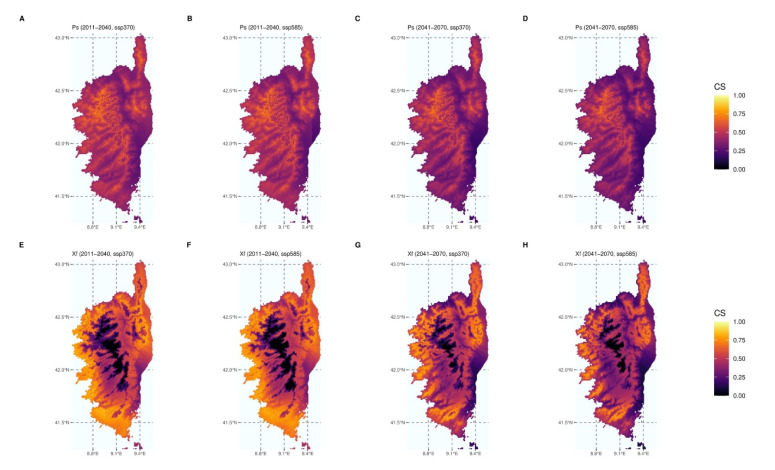
**Species distribution modelling.** Consensus models showing climate suitability (CS) for *P. spumarius* (Ps, (**A**–**D**)) and *Xf* ssp. *multiplex* (Xf, (**E**–**H**)) according to two time periods (2011–2040 and 2041–2070) and two shared socio-economic pathways (SSP370 and SSP585). Climate suitability is estimated using the Maxent algorithm and climate reference data corresponding to the period of 1981–2010.

## Data Availability

Raw sequence reads generated as part of this study are available from the NCBI Sequence Read Archive under BioProject accession PRJNA873902. The code for the analysis of the sequence data is available from github: https://github.com/acruaud/prevalenceXfinsectclimate_2022 (accessed on 12 July 2022).

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
