# Peer review of "Vectors as Sentinels: Rising Temperatures Increase the Risk of Xylella fastidiosa Outbreaks"

_biology, 2022, doi:10.3390/biology11091299_

Round 1
Reviewer 1 Report
The authors analyzed natural infectivity of the plant pathogen, Xylella fastidiosa, in populations of the cosmopolitan vector, Philaenus spumarius, among locations throughout Corsica that differ climatically. They then developed species distribution models for each species, and projected the likely effects of climate change on future occurrence of both species. This topic is of general interest to ecologists and epidemiologists, the study appears to use appropriate methods, and the manuscript is well written. Below are a handful of general suggestions for further strengthening the manuscript.
Currently, the manuscript scarcely describes the criteria used for site selection, which is important for assessing the rigor of such descriptive field studies. Clearly, sites were selected based on expected climate differences, though how they were identified is not clear. Additional details on the sampling habitat type(s), surrounding plant community composition, or other landscape features would help the reader assess whether any potential confounding effects may exist on vector infectivity – especially for a generalist plant pathogen like Xfm.
To what extent might the relatively recent detection of Xfm affect the conclusions of the study? Given that it has apparently been known to be present since only 2015, might that bias estimates of natural infectivity? For example, of the 28 total sampling locations, 20 were sampled in a single year, of which 16 had no apparent Xfm in vector populations. Has the pathogen just not gotten to those locations yet? Again, more details on sight selection could help, if say the authors knew that Xfm was present at all sights albeit at very low prevalence in some of them.
The authors do a nice job with the analysis of the field data, with both models showing significant effects associated with climatic variables. However, they don’t really delve into the significant differences in estimated prevalence among years. Given that the GLMMs already captured interannual differences in climate variables, are those differences among years simply a sampling artifact (maybe more sampling in some years) or might they be explained but some other unaccounted for variable that differs among years?
Given the ambitious scope of the project, the authors assembled an extensive set of appendices that provide important information. However, arguably, there are details buried in the appendices whose inclusion in the manuscript would save the reader some effort. This includes a brief description of the SAFRAN model to clarify that it is spatially interpolated temp/precip estimates updated on a regular basis instead of being historic averages, mention of how much variation PC1 and 2 capture (PC1 is far more important), and the significant effects and coefficients from the GLMMs.
Finally, the SDM projections show estimated suitability for the spittlebug vector and Xfm under different warming scenarios and time horizons. For vector-borne disease, spatial overlap in vector and pathogen occurrence is a useful proxy for disease risk, which is hard to assess given that the authors present the two species side by side. I encourage the authors to consider ways of showing vector/Xfm overlap, which could be addressed by defining threshold suitability values for each species based on model fit metrics, then projecting the areas “suitable” for Ps only, Xfm only, or both.
Reviewer 2 Report
The manuscript submitted by Farigoule et al.provides new and useful information on Xylella fastidiosa vector, Philaenus spumarius, biology and epidemiology in Corsica (France).
A relatively new technique has been used to detect the occurrence of X. fastidiosa subsp. multiplex in the insects. Since the applied technique is not the commonly used for the detection and identification at subspecies level of the pathogen, a more exhaustive explanation of the obtained results concerning the detection of the bacterium is necessary.
The Authors should compare their results with those usually provide by the other techniques utilized for the detection and provide a discussion.
